# Catalytic flexibility of rice glycosyltransferase OsUGT91C1 for the production of palatable steviol glycosides

Jinzhu Zhang[1,5], Minghai Tang[1,5], Yujie Chen[1,5], Dan Ke[1], Jie Zhou[1], Xinyu Xu[1], Wenxian Yang[1], Jianxiong He[1], Haohao Dong [1], Yuquan Wei[1], James H. Naismith [1,2,3], Yi Lin[4], Xiaofeng Zhu [1,6✉] & Wei Cheng [1,6✉]

Steviol glycosides are the intensely sweet components of extracts from *Stevia rebaudiana*. These molecules comprise an invariant steviol aglycone decorated with variable glycans and could widely serve as a low-calorie sweetener. However, the most desirable steviol glycosides Reb D and Reb M, devoid of unpleasant aftertaste, are naturally produced only in trace amounts due to low levels of specific β (1–2) glucosylation in Stevia. Here, we report the biochemical and structural characterization of OsUGT91C1, a glycosyltransferase from *Oryza sativa*, which is efficient at catalyzing β (1–2) glucosylation. The enzyme's ability to bind steviol glycoside substrate in three modes underlies its flexibility to catalyze β (1–2) glucosylation in two distinct orientations as well as β (1–6) glucosylation. Guided by the structural insights, we engineer this enzyme to enhance the desirable β (1–2) glucosylation, eliminate β (1–6) glucosylation, and obtain a promising catalyst for the industrial production of naturally rare but palatable steviol glycosides.

[1] Key Laboratory of Bio-Resource and Eco-Environment of Ministry of Education, College of Life Sciences, State Key Laboratory of Biotherapy and Cancer Center, West China Hospital of Sichuan University, Sichuan University, 610065 Chengdu, China. [2] Division of Structural Biology, Wellcome Trust Centre of Human Genomics, Roosevelt Drive, Oxford OX3 7BN, UK. [3] Rosalind Franklin Institute, Didcot, Oxon OX11 0QS, UK. [4] International Peace Maternity and Child Health Hospital, Institute of Embryo-Fetal Original Adult Disease, School of Medicine, Shanghai Jiao Tong University, 200030 Shanghai, China. [5] These authors contributed equally: Jinzhu Zhang, Minghai Tang, Yujie Chen. [6] These authors jointly supervised this work: Xiaofeng Zhu, Wei Cheng. ✉email: zhuxiaofeng@scu.edu.cn; chengwei669@scu.edu.cn

In addition to serving as an energy source, sugars act as sweeteners that stimulate pleasure neurotransmitters. It is unlikely that humans will abandon the consumption of sweet foods[1]. However, the excessive intake of high-calorie sugars such as glucose, fructose, and sucrose is associated with obesity, diabetes, high blood pressure, and cancer[2,3]. Therefore, public health would benefit substantially from reducing the dietary intake of high-calorie sugars, which can be achieved through the increasing adoption of low-calorie sweeteners that retain the appeal of sweetness with lower risks to health[4].

Steviol glycosides are extracts from the South American shrub *S. rebaudiana* Bertoni, which are ~200 times sweeter than sucrose but have negligible calorific content. They have been used as sweeteners by local populations for centuries[5]. The individual steviol glycoside species comprise an invariant diterpenoid steviol aglycone and two variable glycans. One glycan is attached to the C13-hydroxyl, and the other to the C19-carboxylate of the steviol aglycone[5]. The aglycone is an elongated molecule. The C13-hydroxyl and C19-carboxylate are at two ends of the long axis of the molecule (Fig. 1a). In the previous studies[6], the C13-hydroxyl was termed the R1 end and C19-carboxylate the R2 end. We thus use this simplified nomenclature for consistency.

Steviol glycosides contain a range of compounds. Their glycosylation patterns are variable at both the R1 and R2 ends. The precise chemical structure of the glycans determines sweetness potency and, crucially, the flavor characteristics of the sweet taste (organoleptic properties)[7]. The most naturally abundant steviol glycoside species, stevioside (ST), and rebaudioside A (Reb A)[8,9] are highly sweet (Fig. 1a) but are perceived by some to have a bitter aftertaste, limiting their appeal and hindering their wide adoption[7]. Rebaudioside D (Reb D) and M (Reb M) (Fig. 1a) are intensely sweet and are generally perceived to have excellent organoleptic properties[10,11], an essential combination for more extensive use. However, greatly increased use is not currently practical because both Reb D and Reb M are found only in trace amounts in steviol extracts[10,11].

The glucosylation of steviol species relies on four glycosyltransferases (UGTs) (Fig. 1b) located in the cytosol of *S. rebaudiana*[12–14]. They are all annotated as GT1 family members in the Carbohydrate-Active enZymes (CAZy) Database (www.cazy.org)[15] and utilize uridine diphosphate-activated glucose (UDP-glucose) as the sugar donor. In *S. rebaudiana*, the enzyme UGT85C2 adds the first glucose to the C13-hydroxyl group (R1 end), while UGT74G1 adds the first glucose to the C19-carboxylate moiety (R2 end)[12] (Fig. 1b). These glucose moieties are denoted as glucose 1-R1 (first glucose attached to the R1 end of the aglycone) and glucose 1-R2 (first glucose attached to the R2 end of the aglycone) (Fig. 1a, b). They provide acceptor sites for all subsequent glucose additions at the R1 and R2 ends[16] (Fig. 1a, b). UGT91D2 transfers glucose to make a β (1–2) glycosidic bond with the 2-hydroxyl of glucose 1-R1 (Fig. 1b)[14]. The newly added glucose is thus named glucose 2–1-R1 (glucose linked at the 2-hydroxyl of glucose 1-R1). UGT76G1 transfers glucose to make a β (1–3) glycosidic bond with the 3-hydroxyl of glucose 1-R1 (Fig. 1b). The newly added sugar is termed glucose 3–1-R1. UGT76G1 also transfers glucose to the R2 end to make a β (1–3) glycosidic bond with glucose 1-R2 (Fig. 1a, b). The newly added sugar is thus termed glucose 3–1-R2 (Fig. 1a). The addition of glucose 3–1-R2 is efficient only when glucose 2–1-R2 has been installed[6,14].

Compared to the most abundant steviol glycosides that retain an unpleasant taste, β (1–2) glucosylation at the R2 end (i.e., glucose 2–1-R2) in Reb D and Reb M is essential for the absence of bitter notes in their flavor profiles. However, native UGT91D2 has been reported to catalyze β (1–2) glucosylation almost exclusively at the R1 end but is deficient at the R2 end[14]. The

extremely limited addition of glucose 2-1-R2 by UGT91D2 is therefore highly likely to be responsible for the trace production of Reb D and Reb M in the native plant. Therefore, an efficient enzymatic route to the installation of glucose 2-1-R2 is needed to convert the dominant ST into Reb E and Reb A into Reb D. A glycosyltransferase of *O. sativa* (rice), OsUGT91C1 (originally called EUGT11)[17] was recently identified, which shares 40% identity (56% similarity) to UGT91D2 and shows potential for adding glucose 2-1-R2 to steviol glycosides. UGT76G1 then could follow as a steviol β (1–3) glycosyltransferase to ultimately produce Reb D and Reb M (Fig. 1b).

In this work, we show the biochemical and structural characterization of OsUGT91C1[17] in synthesizing steviol glycosides. OsUGT91C1 is able to catalyze the addition of both glucose 2-1-R1 and glucose 2-1-R2 with roughly equal efficiency. It also catalyzes a third reaction, β (1–6) glucose addition, to create previously uncharacterized compounds. Structural analysis reveals molecular rationales for its catalytic promiscuity and guides the modification of the enzyme to eliminate the unwanted β (1–6) glucosylation and enhance β (1–2) glucosylation on steviol glycoside substrates. The modified enzyme would synthesize more desirable products from the less desirable but abundant steviol glycosides, leading to considerably enhanced yields of Reb D and Reb M.

## Results

**OsUGT91C1 installs both glucose 2-1-R1 and 2-1-R2.** OsUGT91C activity was evaluated by liquid chromatography-mass spectrometry (LC-MS) with UDP-glucose as the donor and commercially available potential substrates as acceptors. Simple glucose is not a substrate for OsUGT91C1, suggesting that the steviol aglycone is required for glucosylation. Rubusoside (Rubu), which contains both glucose 1-R1 and glucose 1-R2 (Fig. 1a), was incubated with the enzyme and UDP-glucose and then sampled at 10, 20, and 30 min (Fig. 2a). Two new UV peaks of a similar-sized area were detected and determined to have the same mass of 803 Da (Fig. 2b, c), corresponding to the addition of 162 Da glucose to Rubu. The peak with a longer retention time was confirmed to be ST (Fig. 1a) using an authentic standard, which arose via the addition of glucose 2-1-R1 to the R1 end of Rubu (Fig. 2e). We termed the earlier eluting product Rub-X. Although no authentic standard was available for Rub-X, tandem MS/MS fragmentation analyses[6] identified two ester-linked glucose units at the R2 end of Rub-X (Fig. 2c).

A five-fold increase in the enzyme concentration led to the detection of a third product. This product was confirmed, by comparison with an authentic standard (Fig. 2a, d), to be Reb E (Fig. 1a). The synthesis of Reb E established that OsUGT91C1 catalyzes β (1–2) glucosylation to both glucose 1-R1 and glucose 1-R2 (Fig. 2e). Therefore, we identified Rub-X as the product resulting from the addition of glucose 2-1-R2 to Rubu. Since the relative proportions of ST and Rub-X remained similar over two hours of turnover (Fig. 2a), we inferred that OsUGT91C1 exhibits no clear preference for the first β (1–2) glucosylation at the R1 or R2 end. After 18 h of incubation, where Reb E was now dominant, there was less Rub-X (Fig. 2a). This required that the conversion of Rub-X to Reb E by the enzyme is more rapid than the conversion of ST to Reb E, suggesting a preference of the R1 end for the second β (1–2) glucosylation.

Reb A, which has a tri-saccharide at the R1 end and glucose 1-R2 at the other end (Fig. 1a), would be expected only to accept a single β (1–2) glucose at the R2 end. Indeed, when Reb A (Fig. 1a) was incubated with OsUGT91C1, it was converted to Reb D by adding glucose 2-1-R2 (Supplementary Fig. 1). Rebaudioside I (Reb I) is related to Reb A but has an additional

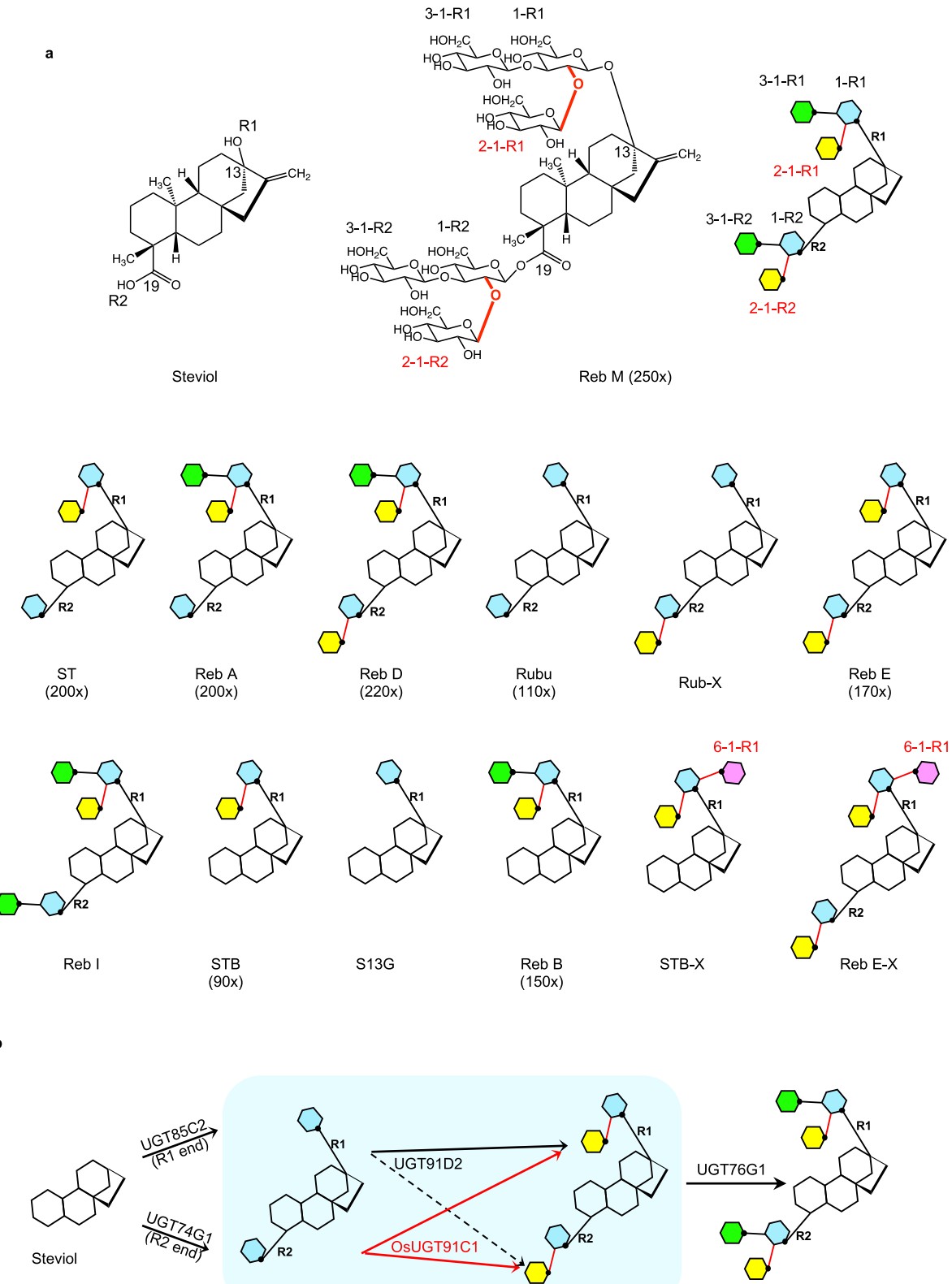

glucose 3-1-R2 at the R2 end (Fig. 1a). We found that Reb I is not a substrate of OsUGT91C1 despite having an acceptor site of glucose 2-1-R2 as Reb A, which suggested that the additional glucose 3-1-R2 appears to prevent the enzyme from adding glucose 2-1-R2. Thus, using this enzyme in synthesizing Reb M from Reb A would have to follow a strict order: β (1–2)

glucosylation by OsUGT91C1, followed by β (1–3) glucosylation by UGT76G1.

**Overall structure of OsUGT91C1.** Based on our above findings that OsUGT91C1 could add glucose 2-1-R1 and 2-1-R2 to both ends of the steviol glycoside substrates, we sought to determine

**Fig. 1 Steviol glucosides related to OsUGT91C1 and the catalysis of the UGTs in synthesizing steviol glucosides. a** Chemical structure of steviol, Reb M, and cartoon representations of the steviol glucoside species. The glycan units are represented by individually colored hexagons with glucose 1-R1 and 1-R2 in cyan, glucose 2-1-R1 and 2-1-R2 in yellow, glucose 3-1-R1 and 3-1-R2 in green, and glucose 6-1-R1 in pink. The anomeric hydroxyl is marked by a black dot. The glycosidic bonds marked in red can be formed by OsUGT91C1. Where known, the sweetness potency of the steviol glycoside sweetener to sucrose is indicated by the number in the bracket. Reb D and Reb M have desirable taste properties. **b** Reactions catalyzed by UGTs in the steviol glycoside biosynthesis pathway. There is no obligate order of the first glucose additions by UGT85C2 (to the R1 end) and UGT74G1 (to the R2 end). The subsequent addition of glucose, the subject of this study, is boxed in blue. UGT91D2 adds glucose 2-1-R1 but shows only trace catalytic activity for the addition of glucose 2-1-R2 (denoted as a dashed arrow). OsUGT91C1, studied here, efficiently adds both glucose 2-1-R1 and glucose 2-1-R2 (as red arrows). UGT76G1 has been previously studied, which adds both glucose 3-1-R1 and glucose 3-1-R2.

what characteristics of its active site enabled this promiscuity. We determined the crystal structure of OsUGT91C1 in (1) the apo form, (2) complex with UDP and Reb E, (3) with UDP and ST, (4) with UDP and STB, and (5) the H27A mutant in complex with UDP and Reb D (Supplementary Table 1 and Supplementary Fig. 2). The apo structure was solved using selenomethionine-labeled OsUGT91C1 and single-wavelength anomalous diffraction. All five structures of OsUGT91C1 share the space group $P2_12_12_1$ with one monomer in the asymmetric unit. PISA analysis[18] indicated no higher-order oligomer in the crystals, and gel filtration suggested OsUGT91C1 remains monomeric in solution.

OsUGT91C1 consists of two Rossmann-like domains (β/α/β) at the N- and C-termini, characteristic of a typical GT-B fold glycosyltransferase[19,20]. The N-terminal domain has seven β strands (Nβ1–7), and the C-terminal domain has six β strands (Cβ1–6). In each Rossmann-like domain, these β strands form a central parallel β-sheet that is sandwiched by the surrounding α-helices and loops (Fig. 3a). Residues 162–201, which connect two helices between β-strands Nβ5 and Nβ6, and residues at the very ends of the N- and C-termini are disordered in all the structures. Structural comparisons of the four complex structures showed that the protein structure is essentially unchanged (r.m.s.d. of 0.4 Å) regardless of which steviol compound is bound. However, a comparison between the complex structures and the apo structure showed that the C-terminal Rossmann domain undergoes conformational change (r.m.s.d. of 2.3 Å), most likely due to the binding of UDP (Supplementary Fig. 3).

UDP is bound identically in all four complexes and surrounded by the helices Nα1, Cα4, and the loops at the C-terminal side of β-strands Cβ1, Cβ3, and Cβ4 (Fig. 3b and Supplementary Fig. 4). The uridine ring of UDP stacks with Trp339 and forms two hydrogen bonds with the amide and carbonyl groups of Val340 (Fig. 3b). The 2- and 3-hydroxyl groups of the ribose ring form a bidentate hydrogen bond with Glu365, which is in turn hydrogen-bonded to Gln342 and Arg255 (Fig. 3b). Arg255 also forms a hydrogen bond to the 2-hydroxyl of the ribose. The pyrophosphate of UDP is hydrogen-bonded to Ser282, His357, Asn361, and Ser362 from the C-terminal domain (Fig. 3b).

The β-phosphate of UDP (to which glucose would be attached) is also hydrogen-bonded to a cluster of waters (or to a glycerol molecule in some structures) (Fig. 3b and Supplementary Fig. 4). The volume occupied by the water molecules (or glycerol) represents the most likely position of glucose in a UDP-glucose molecule[21]. The cluster of water molecules (or glycerol) forms hydrogen bonds with the side chains of Asp381, Gln382, and Trp360 (Supplementary Fig. 4).

In complex structures, the steviol glycoside substrates bind in two different orientations. In one orientation, the R2 end is at the active site with the R1 end at the "out" site on the protein surface (Fig. 3c, d), and in the other orientation, the steviol aglycone rotates 180° around an axis perpendicular to the center of the aglycone, such that the R1 end is at the active site with the R2 end at the "out" site (Fig. 3e). Two opposite binding orientations

demonstrated that both ends of the steviol glycoside substrates are able to enter the active site for β (1–2) glucosylation.

**Substrate recognition with the R2 end at the active site.** Three complexes, native OsUGT91C1 with UDP and Reb E, native OsUGT91C1 with UDP and ST, and the H27A mutant with UDP and Reb D, bind steviol compounds with the R2 end at the active site. The positions of the common atoms of Reb E (Fig. 3c) and ST (Fig. 3d) are essentially identical, and we describe the higher-resolution Reb E complex in detail (Fig. 3c) about the catalytic mechanism and substrate recognition mode of OsUGT91C1 for β (1–2) glucosylation at the R2 end.

The 2-hydroxyl of glucose 1-R2 is hydrogen-bonded to His27 (Fig. 3c). His27 is, in turn, hydrogen-bonded to Asp128, creating a catalytic dyad, one of the catalytic motifs employed by the inverting GT-B glycosyltransferase[19]. In this mechanism, His27 acts as the general base in a relay with Asp128 to deprotonate the 2-hydroxyl of the accepting glucose at the active site, creating the nucleophile for the $S_N2$ reaction with UDP-glucose (Supplementary Fig. 5e)[22]. The mutation of His27 to Ala completely inactivates OsUGT91C1 in β (1–2) glucosylation at either end of any previously assayed substrate (Supplementary Fig. 5a, b). The inactivity of H27A and the single binding site for UDP leads to the conclusion that there is a single active site in the enzyme that catalyzes both glucose 2-1-R1 and 2-1-R2 addition.

At the active site, glucose 1-R2 is held by Phe379, His93, Trp22, His27, Phe130, and Leu204. The 3- and 4-hydroxyls of glucose 1-R2 form a bidentate hydrogen bond with Glu283 (Fig. 3c, d). Mutation of Glu283 to either Gln or Ala significantly decreases enzyme activity (Supplementary Fig. 5c, d), confirming that the bidentate hydrogen bond to Glu283 is critical for recognition. The 3-hydroxyl of glucose 1-R2 is hydrogen-bonded to Trp22 (Fig. 3c, d); thus, any sugar attachment to this 3-hydroxyl would result in severe steric clashes. This observation rationalized the lack of turnover of Reb I, which has glucose at the 3-hydroxyl of glucose 1-R2 (Fig. 1a). We did not identify any convincing density in the Reb E complex for its glucose 2-1-R2, indicating that the sugar was cleaved during crystallization or was disordered (Fig. 3c). We do not favor the disorder of glucose 2-1-R1 as an explanation since the ordered water molecules and surrounding residues occupy the space to preclude the presence of glucose 2-1-R2.

The steviol aglycone of the substrate sits in a cleft formed by Val129, Phe130, Leu149, Met155, Ile159, Arg162, Ala205, and Phe208 (Fig. 3c, d). Notably, none of these residues interact specifically with the aglycone. Consistent with this rather nonspecific interaction, comparing the three complexes shows shifts between the aglycone atoms relative to the protein (Supplementary Fig. 6a).

The R1 end of the bound molecule is at the "out" site, close to the common disordered region between residues 163 and 201 (Fig. 3c, d). The R1 glycan of ST was not modeled into the weak experimental electron density observed at this location, indicating disorder due to flexibility (Fig. 3d and Supplementary Fig. 2b).

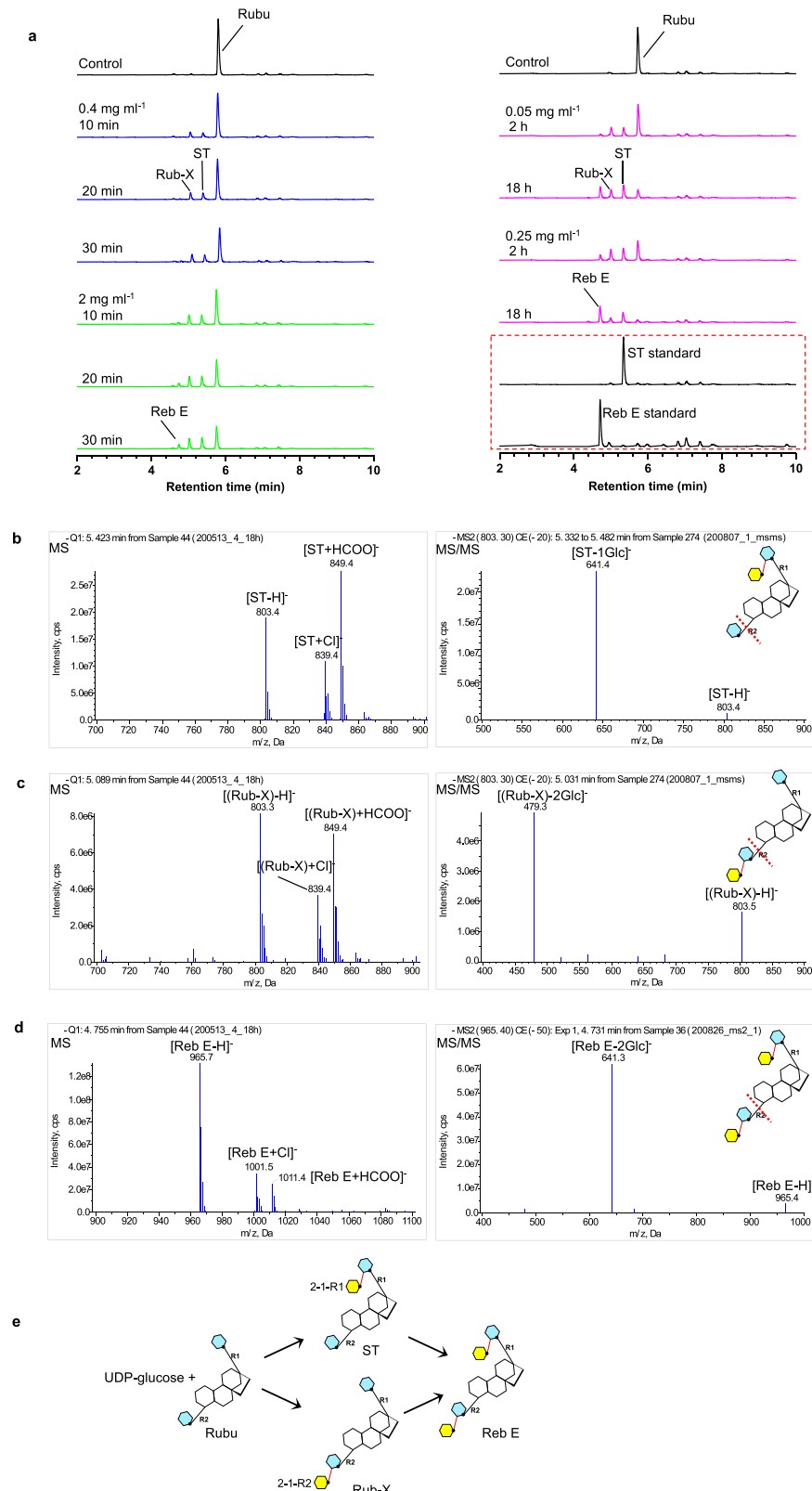

The trisaccharide attached to the R1 end of Reb D was built into a weak but unambiguous density (Supplementary Fig. 2c). Reb E should have a di-saccharide at the R1 end (Fig. 1a), but the electron density clearly shows the presence of a tri-saccharide (Supplementary Fig. 2a). We modeled the third sugar as β (1–6) glucose linked to the 6-hydroxyl of glucose 1-R1 (Fig. 3c and

Supplementary Fig. 2a). This would imply that additional glucosylation at the R1 end occurred during the crystallization process.

**Substrate recognition with the R1 end at the active site**. We chose STB (Fig. 1a), which lacks the R2 glycan, as a means to

**Fig. 2 β (1–2) sugar transfers by OsUGT91C1 at both the R1 and R2 ends of Rubu. a** LC-MS was used to monitor the reaction of Rubu with OsUGT91C1 and UDP-glucose. The HPLC traces represent the 18 h control reaction without the enzyme (black); the incubations for 10 min, 20 min, and 30 min with the enzyme at 0.4 mg ml$^{-1}$ (blue); the incubations were repeated with the enzyme (5x) at 2.0 mg ml$^{-1}$ (green). The HPLC traces in pink report the incubation with the enzyme at 0.05 mg ml$^{-1}$ for 2 h and 18 h, repeated with 0.25 mg ml$^{-1}$ enzyme (5x). The authentic standards ST and Reb E are shown in the red dashed box. **b–d** Mass spectra of the three new peaks in LC-MS are consistent with the assignment of ST (**b**), Rub-X (**c**), and Reb E (**d**). The main negative derived ions of the products ST, Rub-X, and Reb E are labeled in MS analyses. The negative parent ions [ST-H]$^{-}$, [(Rub-X)-H]$^{-}$, and [Reb E-H]$^{-}$ with $m/z$ at 803, 803, and 965 were explicitly isolated and then characterized by MS/MS, where the more labile ester bond breaks first in MS/MS, leading to the first mass loss of ester-linked glycan at the R2 end from the parent ion. The size of the mass loss of the abundant fragment ions of ST, Rub-X, and Reb E indicates the number of ester-linked glucose units at the R2 end. The insert denotes where the ester bond breaks first during MS/MS fragmentation with a red dash line. **e** Cartoon of the reactions catalyzed by OsUGT91C1 on Rubu.

obtain a complex with the R1 end at the catalytic site (Fig. 3e). We could build both glucose 1-R1 and glucose 2-1-R1 into the electron density at the active site (Supplementary Fig. 2d). Compared to the complexes with the R2 end at the catalytic site, the steviol aglycone of STB has undergone a 180° rotation to swap the R1 and R2 ends (Fig. 3e, f). The position and volume occupied by the steviol aglycone in this rotated arrangement are very similar to those complexes with R2 glycans at the catalytic site (Fig. 3f). Moreover, the spatial arrangement of the catalytic residues is unchanged (Fig. 3c–e). If we applied the same 180° rotation of steviol aglycone to the Reb E complex, it would move glucose 1-R1 from the "out" site to the catalytic site, precisely swapping the original position, geometry, and binding environment between glucose 1-R1 and 1-R2. As a result, the 2-hydroxyl of glucose 1-R1 is within the hydrogen bond distance to His27 and thus ready for catalysis, providing the substrate recognition mode for β (1–2) glucosylation at the R1 end.

In addition to steviol aglycone rotation, glucose 1-R1 of STB undergoes a 180° flip around the glycosidic bond to the aglycone (Fig. 3e, f). The same flip transition of glucose 1-R1 was also observed between the R1 tri-saccharides of Reb D and Reb E (Supplementary Fig. 6b). As a result of the flip, it is 6-hydroxyl, not 2-hydroxyl of glucose 1-R1, hydrogen-bonded to His27 (Fig. 3e). In this orientation, glucose 1-R1 does not form a bidentate hydrogen bond with Glu283 but retains the hydrogen bonds of the 4-hydroxyl with Glu283 and Trp22 (Fig. 3e). Glucose 2-1-R1 is found in a previously undisclosed pocket where it makes contact with His93, and its hydrophobic face stacks against the aromatic ring of Phe379 (Fig. 3e). We believe that these interactions with glucose 2-1-R1 stabilize the flipped arrangement of glucose 1-R1 (Fig. 3e).

**OsUGT91C1 catalyzes additional reactions**. The Reb E complex indicated that glucose 2-1-R2 had been removed. To evaluate whether the cleavage of this β (1–2) glycosidic bond results from enzyme activity, we incubated OsUGT91C1 with Reb E and UDP in solution. LC-MS confirmed that Reb E was converted to both ST and Rub-X, and eventually, these compounds were transformed to Rubu (Supplementary Fig. 7a). Incubation of ST with UDP and enzyme produced Rubu, incubation with STB produced S13G, and incubation with Reb D produced Reb A (Supplementary Fig. 7b–d). The H27A mutant is inactive (Supplementary Fig. 7e), confirming that the cleavage arises from the same catalytic machinery of β (1–2) glucosylation. We managed to obtain the structure of the H27A inactivated mutant in a complex with Reb D and UDP (Supplementary Fig. 7f). The previously invisible glucose 2-1-R2 is well-ordered. We have not tried to detect the production of UDP-glucose but suspect that the removal of sugar is simply the reverse reaction of glucosylation (sugar transfer back to UDP) that has been seen in other glycosyltransferases[23–25].

The Reb E structure suggested the presence of glucose 6-1-R1 (Fig. 3c), and the complex with STB showed that 6-hydroxyl of

glucose 1-R1 is hydrogen-bonded to the catalytic His27 (Fig. 3e). Incubation of the enzyme and UDP-glucose with STB (Fig. 4a–c) and with Reb E (Fig. 4d–f) showed in each case a single glucose addition to the R1 end. We termed these compounds STB-X and Reb E-X. The H27A mutant deteriorates in reaction, confirming that it utilizes the same active site (Supplementary Fig. 8). Based on the crystal structures, we identified this activity as β (1–6) glucosylation (Fig. 4c, f). Reb D has additional glucose 3-1-R1 than Reb E, which was not modified by the enzyme. Similar to β (1–2) glucosylation, glucose 3-1-R1 blocks β (1–6) glucosylation. This β (1–6) modification was not present in steviol sweeteners. As a result, compounds with β (1–6) glucosylation remain untested for sweetness or organoleptic properties and may represent a potentially undesirable feature.

**Engineering of OsUGT91C1**. Rubu (modifiable at both the R1 and R2 ends), Reb A (only modifiable at the R2 end), and S13G (only modifiable at the R1 end) (Fig. 1a) were used as test substrates. We set out to enhance β (1–2) glucosylation, particularly at the R2 end. We reasoned that creating a more flexible steviol binding site would allow the enzyme to recognize both orientations of the aglycone more readily. We chose the mutant F208M to preserve hydrophobicity while creating more opportunities for the protein and aglycone to adjust their fit. This mutant demonstrated a preference for the R2 sugar addition compared to the native enzyme that showed no preference (Supplementary Fig. 9). The F208M mutant showed a 4-fold increase in $k_{cat}/K_m$ (Table 1) for Reb A (R2 addition), while with S13G (R1 addition), an around 2-fold increase in $k_{cat}/K_m$ was observed (Table 1). Rubu (R1 and R2 additions) showed a nearly 3-fold improvement in $k_{cat}/K_m$ (Table 1). Thus, this mutation accelerated the catalytic efficiency of β (1–2) glucosylation at both ends with a preference for the R2 end.

Guided by structure, we constructed three mutants H93A, H93W, and F379A, to suppress β (1–6) glucosylation by eliminating the cryptic glucose pocket seen for glucose 2-1-R1 in the STB complex (Fig. 3e). H93A decreased but did not eliminate the side reaction of β (1–6) glucosylation (Supplementary Fig. 10b). H93W eliminated β (1–6) glucosylation activity (Supplementary Fig. 10c) and also decreased β (1–2) glucosylation with both Reb A and Rubu (Table 1). F379A eliminated β (1–6) activity (Supplementary Fig. 10e) but improved the β (1–2) reaction significantly (Table 1).

The double mutants H93W/F208M and F379A/F208M were tested, and both suppressed β (1–6) glucosylation activity (Supplementary Fig. 10g, h). H93W/F208M showed the same $k_{cat}/K_m$ for Rubu, S13G, and Reb A as the wild-type enzyme. F379A/F208M showed a 3-fold increase in $k_{cat}/K_m$ for S13G, a 4-fold increase in $k_{cat}/K_m$ for Reb A, and an almost 6-fold increase in $k_{cat}/K_m$ for Rubu with a preferred addition at the R2 end (Table 1). The double mutant F379A/F208M seems well suited to further exploitation.

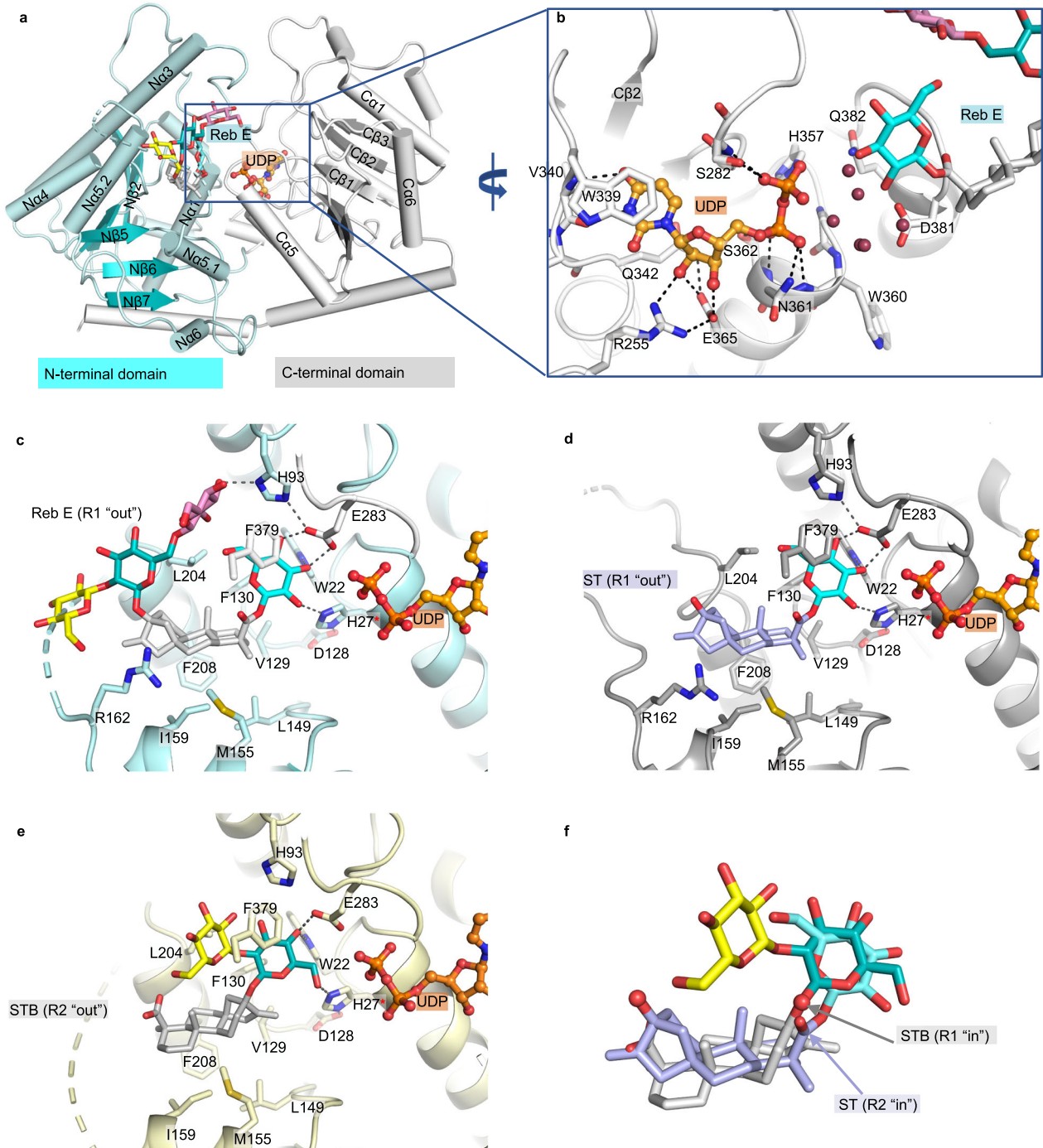

**Fig. 3 Structure of OsUGT91C1 and the binding modes to recognize substrate. a** Overall structure of OsUGT91C1. The structure is shown in cartoon representation with the N-terminus domain colored cyan and the C-terminus white. The bound UDP is shown in stick-ball mode with carbon colored orange, oxygen red, and nitrogen blue. The steviol compound derived from Reb E is shown in stick with carbons of the steviol aglycone colored white, while carbons of the glycan are colored differently, carbons of glucose 1-R2 in cyan, glucose 1-R1 in dark cyan, glucose 2-1-R1 in yellow, and glucose 6-1-R1 in pink. **b** Binding of UDP mainly involves the C-terminal domain with the hydrogen bonds shown by dashed lines. A cluster of water molecules, shown as red balls, fills the space most likely occupied by the glucose of UDP-glucose. **c**, **d** Binding of the steviol compounds Reb E (**c**) and ST (**d**) with the R2 end at the active site and the R1 end at the "out" site. **e** Binding of the steviol compound STB with the R1 end at the active site and the R2 end at the "out" site. The catalytic His27 is marked with a red asterisk in **c–e**. **f** Comparison of two binding modes using ST (the R2 end at the active site, denoted as R2 "in") and STB (the R1 end at the active site, denoted as R1 "in").

## Discussion

Steviol glycosides are a class of low-calorie sweet compounds. They are formed by glucosylation on both the R1 and R2 ends of the steviol aglycone. The assembly of the glycans requires several glycosyltransferases (Fig. 1b). The endogenous UGT91D2 in *S.* *rebaudiana* is highly specific to catalyze β (1–2) glucosylation to the R1 end but shows very little catalysis to the R2 end. This is a drawback as β (1–2) glucosylation to the R2 end is desirable to create more valuable steviol glycosides. OsUGT91C1 from rice is a homolog of UGT91D2 and has a more relaxed specificity, which

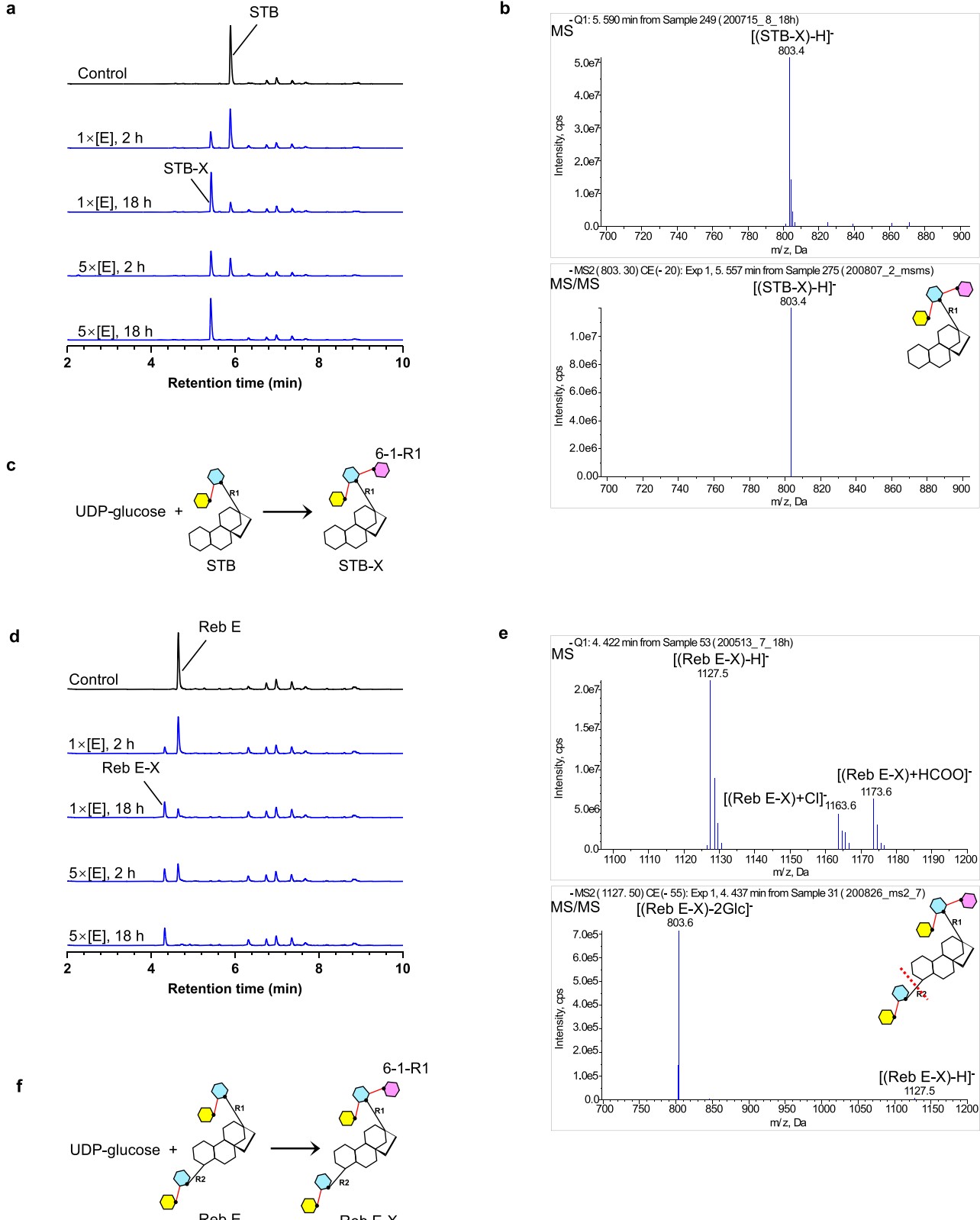

**Fig. 4 β (1–6) sugar transfer by OsUGT91C1 at the R1 end of steviol substrates STB and Reb E. a**, **d** LC-MS was used to monitor the reaction progress of STB (**a**) or Reb E (**d**) with OsUGT91C1. The HPLC traces represent the 18 h control reaction without the enzyme (black); the incubations for 2 h and 18 h with the enzyme at 0.15 mg ml⁻¹; repeated with the enzyme (5x) at 0.75 mg ml⁻¹ (blue). The yields of the products are related to the enzyme concentration and the reaction duration, consistent with enzymatic-catalyzed reaction. **b**, **e** Mass spectra of the new peak in LC-MS are consistent with a single glucose mass addition at the R1 end of the molecule. The products are arbitrarily named STB-X (**b**) and Reb E-X (**e**). The main negative derived ions are labeled in MS analyses. The negative parent ions [(STB-X)-H]⁻ with $m/z$ at 803 and [(Reb E-X)-H]⁻ with $m/z$ at 1127 were explicitly isolated and characterized by MS/MS. The insert notes where the ester bond breaks first during MS/MS fragmentation with a red dash line. **c**, **f** Reaction schemes catalyzed by OsUGT91C1 on substrate STB (**c**) and Reb E (**f**).

**Table 1 Enzyme kinetic parameters for the sugar transfer of OsUGT91C1 and the mutants.**

| Substrate | Enzyme | $k_{cat}$ (min$^{-1}$) | $K_m$ (µM) | $k_{cat}/K_m$ (s$^{-1}$ M$^{-1}$) | Fold |
|---|---|---|---|---|---|
| Rubu | WT | 2.0 ± 0.1 | 49.5 ± 5.3 | 673 | 1 |
| (Combined β (1–2) | F208M | 2.4 ± 0.1 | 20.9 ± 1.7 | 1914 | 2.8 |
| at R1 and R2) | H93W | 1.3 ± 0.1 | 56.8 ± 8.5 | 381 | 0.6 |
| | F379A | 2.7 ± 0.1 | 16.5 ± 2.8 | 2727 | 4.1 |
| | H93W/F208M | 1.8 ± 0.1 | 43.6 ± 5.4 | 688 | 1.0 |
| | F379A/F208M | 2.8 ± 0.1 | 11.7 ± 1.0 | 3989 | 5.9 |
| S13G | WT | 2.9 ± 0.1 | 24.9 ± 2.5 | 1941 | 1 |
| (β (1–2) at R1) | F208M | 3.9 ± 0.1 | 18.7 ± 1.7 | 3476 | 1.8 |
| | H93W | 3.9 ± 0.1 | 36.2 ± 4.7 | 1796 | 0.9 |
| | F379A | 5.8 ± 0.2 | 19.0 ± 2.6 | 5088 | 2.6 |
| | H93W/F208M | 3.5 ± 0.1 | 29.1 ± 2.2 | 2005 | 1.0 |
| | F379A/F208M | 3.5 ± 0.1 | 9.7 ± 1.9 | 6014 | 3.1 |
| Reb A | WT | 1.22 ± 0.02 | 45.7 ± 3.0 | 445 | 1 |
| (β (1–2) at R2) | F208M | 2.9 ± 0.1 | 25.2 ± 2.0 | 1918 | 4.3 |
| | H93W | 0.60 ± 0.03 | 58 ± 12 | 172 | 0.4 |
| | F379A | 1.16 ± 0.02 | 37.0 ± 2.5 | 523 | 1.2 |
| | H93W/F208M | 2.05 ± 0.03 | 63.9 ± 3.6 | 535 | 1.2 |
| | F379A/F208M | 1.7 ± 0.1 | 15.1 ± 2.6 | 1876 | 4.2 |
| STB | WT | 0.63 ± 0.03 | 71 ± 12 | 148 | – |
| (β (1–6) at R1) | H93W | ND | – | – | – |
| | F379A | ND | – | – | – |
| | H93W/F208M | ND | – | – | – |
| | F379A/F208M | ND | – | – | – |

*WT* wild type, *ND* non-detectable, *fold* fold change over the $k_{cat}/K_m$ of the wide type for each substrate.
Assays were performed as described in Methods section. Nonlinear fitting values ±SD ($n = 3$) are shown. Source data are provided as a Source Data file.

can catalyze β (1–2) glucosylation to both glucose 1-R1 and glucose 1-R2 and potentially transform the abundant steviol glycosides to superior ones with quality taste.

As OsUGT91C1 shows catalytic flexibility, suggesting the lack of stringent substrate recognition[26–28], it is often difficult to obtain structural information on the different substrate recognition modes, hindering promiscuity analysis and insight[29,30]. OsUGT91C1 is a rare case that multiple structures were obtained to describe the substrate in different poses and rationalize promiscuity and regiospecificity directly. In addition, β (1–6) glucosylation would not be noted unless the complex structure with STB and Reb E were available. The steviol glycoside substrate in OsUGT91C1 exhibits two motions for promiscuity, R1/R2 rotational swap of steviol aglycone and planar flip of glucose 1-R1. When these two motions are combined, the steviol aglycone and the linked glucose are concertedly recognized in three distinctive modes so that three specific hydroxyl groups of the steviol glycoside substrate can be delivered towards the activation zone of the catalytic His27, then activated to accept the glucose from the sugar donor (Fig. 5a–c).

OsUGT91C1 creates a rather loose-fitting hydrophobic tunnel that accommodates the steviol aglycone in two orientations by a combination of two factors. First, the steviol aglycone has a relatively flat, featureless elongated structure that exhibits pseudo-two-fold symmetry. Thus, whether the R1 or R2 end is at the active site, the aglycone presents a similar recognition challenge. Second, OsUGT91C1 relies exclusively on nonspecific hydrophobic interactions inside a tunnel to bind the aglycone (Fig. 5a, b). This combination allows hydrophobic binding energy to be captured in two different orientations (the R1 or R2 end at the active site). The aglycone binding pocket in OsUGT91C1 is shallow and widely open (Fig. 5a), which relieves any clashes and allows the active site to position the glucose molecules (glucose 1-R1 or 1-R2) attached to either end of the aglycone equally well. Our analysis of the different poses led to engineering a mutant F208M to enhance the flexible fit of

aglycone, and this mutant showed improved catalytic efficiency, particularly at the R2 end.

UGT76G1 is the other steviol glycosyltransferase with available structures[6]. Both OsUGT91C1 and UGT76G1 share a common fold that belongs to the same CAZy family. However, their structural details are quite different (Fig. 5). UGT76G1 has a deep aglycone pocket formed by different elements of secondary structure (Fig. 5d) than are used by OsUGT91C1 to form its aglycone tunnel (Fig. 5a). As a consequence of the different constructions and locations of the aglycone site, there is an ~60° rotation difference in the position of glucose at the catalytic site (Fig. 5a, d). As a result, 2-hydroxyl of glucose is hydrogen-bonded to the catalytic histidine in OsUGT91C1 (Fig. 5a) and 3-hydroxyl of glucose in UGT76G1 (Fig. 5d), showing β (1–2) and β (1–3) regioselectivity, respectively[66].

We observed that OsUGT91C1 catalyzes a third reaction, β (1–6) glucosylation at the R1 end (Fig. 4). This second example of catalytic promiscuity arises from different structural features that allow catalysis at both the R1 and R2 ends. In OsUGT91C1, there is an additional glucose pocket adjacent to the active site that can bind glucose 2-1-R1 (Figs. 3e and 5c). Thus, after glucose 2-1-R1 is installed, glucose 1-R1 flips around its glycosidic linkage to the aglycone and moves glucose 2-1-R1 out of the active site into the additional glucose pocket (Figs. 3e and 5c). By clearing the active site, OsUGT91C1 can bind a fresh UDP-glucose molecule. The second consequence of glucose 1-R1 flipping is that it places the 6-hydroxyl of glucose 1-R1 in a position for catalysis (Figs. 3e and 5c), switching to β (1–6) after β (1–2) glucosylation. β (1–6) glucosylation is seen only at the R1 end (Fig. 4), and our analysis did identify several subtle differences in the interactions between substrate and protein when the R2 rather than R1 end is at the active site (Fig. 3c–f). We presume that these differences prevent the precise positioning of the 6-hydroxyl of glucose 1-R2 required for catalysis. Guided by structural analysis, we were able to disrupt this additional glucose pocket by mutation (H93W, F379A) and suppress β (1–6) glucosylation (Supplementary Fig. 10).

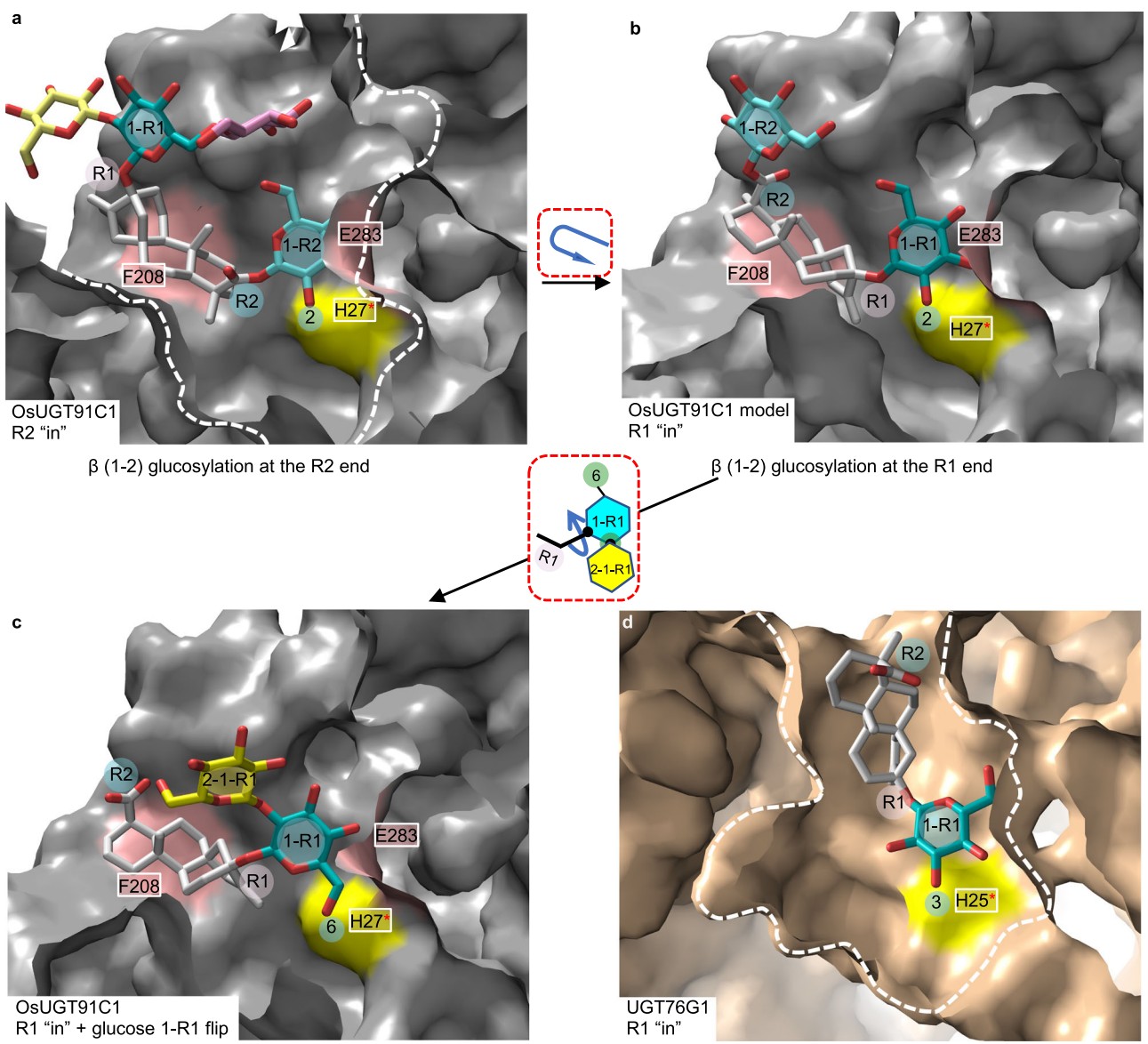

**Fig. 5 Origins of OsUGT91C1 promiscuity. a** OsUGT91C1 binds Reb E with the R2 end at the active site is ready to add glucose 2-1-R2 to glucose 1-R2. The protein is represented in the surface, and the loose-fitting substrate tunnel of OsUGT91C1 is delimited by white dashed lines. The catalytic residue His27 is colored yellow (marked with a red asterisk), and the residues Phe208 and Glu283 are colored salmon on the surface. Phe379 cannot be seen as a result of the cutaway surface presentation, and this residue is shown in Fig. 3c–e. The positions of the R1 and R2 ends, glucose units, the reactive 2-hydroxyl, and 6-hydroxyl are labeled with a circle shadow for clarity. **b** OsUGT91C1 is able to bind the aglycone in the opposite orientation (the transition is shown in the red dashed box) so that the R1 end is now at the active site positioned for the addition of glucose 2-1-R1 to glucose 1-R1. The figure uses the same design as **a**. **c** After the addition of glucose 2-1-R1, glucose 1-R1 has flipped by 180° from that seen in **b** (the transition is shown in the red dashed box). As a result, glucose 2-1-R1 is held in a new pocket, freeing up the enzyme to bind fresh UDP-glucose and positioning the 6-OH of glucose 1-R1 for catalysis. The figure uses the same design as **a**. **d** Structurally related β (1–3) glycosyltransferase UGT76G1 (Fig. 1b) binds the aglycone in a different location. As a result, 3-OH, not 2-OH, is positioned for catalysis. The figure follows the design of **a** with color differences. The orientation of UGT76G1 here matches that of OsUGT91C1 in **a–c**. In UGT76G1, His25 serves as the catalytic base and is colored yellow (marked with a red asterisk) on the surface.

OsUGT91C1 shows unusual catalytic promiscuity in that it catalyzes three distinct glucosylation reactions at the same active site, β (1–2) to the R1 end and R2 end, β (1–6) to the R1 end (Fig. 5a–c). OsUGT91C1 is, therefore, an exception to the "one glycosyltransferase, one specific linkage" rule of thumb[26]. Other exceptions to the rule are known, including the well-studied UDP-Gal:GlcNAcβ-R β1-4 galactosyltransferase (β4-GalT), which can transfer galactose to both the 4-hydroxyl of the terminal glucosamine of a protein glycan and the 4-hydroxyl of free glucose when bound to α-lactalbumin[31]. The UDP-dependent O-glucosyltransferases involved in flavonoid decoration[32,33] are other well-known systems that break the simple rule. Here we have experimentally visualized how the promiscuity of OsUGT91C1 arises from two distinct structural features (the flexible aglycone site and the cryptic glucose pocket) and have demonstrated that we can control the promiscuity to create an engineered OsUGT91C1 that does not add β (1–6) glucose but has improved catalytic efficiency of the desirable β (1–2) glucosylation at the R2 end of steviol glycoside substrates. This engineered enzyme would relieve the catalytic bottleneck that makes only trace amounts of the most desirable products in the native plant. As a result, an

inexpensive, biological source of low-calorie sweeteners with a desirable taste profile would be readily available and have more beneficial impacts on human health.

## Methods

**Protein expression and purification.** The encoding region of OsUGT91C1 of *O. sativa* was *de novo* synthesized and subcloned into the expression vector pET28b. The constructed plasmid was transformed into *Escherichia coli* BL21 (DE3) (Novagen) for overexpression. For native OsUGT91C1, 0.5 mM IPTG was used to induce the expression for 4 h at 37 °C. For selenomethionine (SeMet)-labeled protein, *E. coli* cells were initially incubated in M9 minimal media at 37 °C until the cell density reached 1.0 at 600 nm. SeMet and IPTG were added, and the cells were cultivated for another 18 h at 16 °C. The *E. coli* cells were harvested by centrifugation and then lysed by sonication. The supernatant was loaded on a 5 ml HisTrap affinity column (GE Life Sciences), and His-tagged OsUGT91C1 was eluted with 20 mM Tris-HCl buffer pH 7.8, 0.5 M NaCl, and 250 mM imidazole, followed by immediate gel filtration in 10 mM HEPES-NaOH buffer pH 7.2, 150 mM NaCl, and 2 mM DTT. The protein was finally concentrated to 20 mg ml$^{-1}$ and stored at −80 °C in aliquots. Site-directed mutants of OsUGT91C1 were generated using the Quick-Change PCR mutagenesis protocol.

**Structural biology.** The apo protein (20 mg ml$^{-1}$) or the protein sample mixed with 1 mM UDP and 5 mM of the various steviol glycoside substrates in DMSO solution were crystallized using the sitting-drop vapor-diffusion method. After initial screening and optimizing the crystalization conditions, the diffractive crystals were obtained in 100 mM HEPES-NaOH buffer pH 6.5 and 20% PEG 4000. The crystals were flash-frozen and stored in liquid N$_2$ after soaking in the cryoprotectant, containing 25% glycerol and the original crystalization composition.

X-ray diffraction data were collected at beamline BL18U1 or BL19U1 at Shanghai Synchrotron Radiation Facility (SSRF, National Center for Protein Science Shanghai, Institute of Biochemistry and Cell Biology, Chinese Academy of Sciences, P. R. China) using a Pilatus detector at a wavelength of 0.97853 Å. The data were indexed, integrated, and scaled using DIALS[34] (version 3.0) in the CCP4i2[35] package (version 1.0). The crystals were inclined to deteriorate in the cryoprotectant. Only one crystal of SeMet-labeled OsUGT91C1 survived the cryo-treatment and gave a 3.45 Å single-wavelength anomalous diffraction (SAD) dataset. A partial model was generated with CRANK2[36] (version 2.0) and then used as a molecular replacement search model in Phaser[37] (version 2.8) for a higher-resolution native dataset where the complete model can be built. The apo structure acted as the search model to solve the other complex structures. Manual model building and subsequent refinement were performed using Coot[38] (version 0.8) and Refmac5[39] (version 5.8). The restraint libraries for all the steviol compounds were visually built in Coot and then idealized by ProDrg[40] (version 2.5) and Acedrg[41] (version 217) of the CCP4i package 7.1. The geometry of each steviol glycoside after refinement was validated by Privateer[42] (version MKIII) of the CCP4i2 package (version 1.1). The statistics of data collection and refinement are summarized in Supplementary Table 1.

**Biochemical assays.** The in vitro biochemical assays of OsUGT91C1 and the mutants were performed at least in triplicate. 0.3 mM of each steviol glycoside substrate, including Rubu, Reb A, STB, Reb E, Reb D, and ST, was added to 200 µl of the reaction mix consisting of 20 mM Tris-HCl buffer pH 7.2 and 1 mM UDP-glucose. The reactions were initialized by two different concentrations of purified OsUGT91C1 or the mutants (1× or 5×, the actual protein concentrations are indicated in the figure legends) and incubated at 25 °C. In all, 60 µl of respective aliquots were sampled at 0, 2, and 18 h of reaction and mixed with an equal volume of 1-butanol to terminate the reaction and extract the hydrophobic steviol glycoside substrate and products. After centrifugation at 17,000×*g* for 10 min, the upper butanol layer was collected, evaporated, and resuspended in 25% acetonitrile, which was subjected to liquid chromatography-mass spectrometry (LC-MS) or LC-MS/MS.

LC-MS and LC-MS/MS analyses were performed on a Shimadzu Ultra-fast liquid chromatography system (UFLC, Shimadzu) coupled with an AB SCIEX Qtrap 5500 mass spectrometer, which is equipped with a Turbo Spray ion source. Chromatographic separation was achieved on a Waters ACQUITY UPLC BEH C18 column (2.1 mm × 100 mm I.D., 1.7 µm) by a mobile phase consisting of water and acetonitrile, running at a flow rate of 0.5 ml/min and three elution steps that included 5% acetonitrile for 0–1 min, a linear gradient of 5–95% acetonitrile for 1–10 min, and 95% acetonitrile for 10–12 min. The column and autosampler were maintained at 40 °C and 15 °C, respectively. In all, 1 µl of the sample was injected and detected at a wavelength of 210 nm for steviol compounds and simultaneous MS or MS/MS in negative ionization mode. The mass spectrometric parameters were optimized as follows: ion spray voltage at −4500 V, declustering Potential at −90 V, and the temperature set to 500 °C. Q1 scan was scanned in the range of 200–1200 Da. In the MS/MS product ion mode, the parent ions of steviol glycosides (*m/z* 641.3, *m/z* 803.3, *m/z* 965.4, *m/z* 1127.5) were captured and fragmented under collision energies ranging from 20 to 60 eV. The chromatographic and mass spectrometry data were collected and analyzed in

Analyst 1.6.2 software (AB SCIEX, USA). Authentic steviol glycoside standards were purchased (ChromaDex, USA).

**Steady-state enzyme kinetic assays.** Steady-state kinetic assays of those substrates were performed using the UDP-Glo$^{TM}$ Glycosyltransferase Assay Kit (Promega). The sugar transfer of the glycosyltransferase turns UDP-glucose to UDP, which is converted to ATP and then quantified using a luciferase/luciferin reaction. The luminescence intensity linearly correlates to the concentration of UDP when it is <25 µM, thus reflecting UDP production during the reaction. Skanlt software 2.4.3 RE controlled Varioskan™ Flash multimode microplate reader and recorded the luminescence intensity (Thermo Scientific, USA). A standard linear plot between the luminescence intensity and UDP concentrations was drawn for each microplate measurement ($R^2 > 0.98$).

After the reaction was started by adding the enzyme, four individual aliquots were taken every 1 min and mixed with an equal volume of the UDP detection reagent, terminating the reaction and quantifying the velocity of UDP production. The blank turnover of UDPG to UDP in the absence of the steviol glycoside substrate was subtracted from each velocity calculation. The appropriate concentrations of the enzyme were tested to ensure that the consumption of both steviol glycoside substrate and UDPG was <10% of the initial concentration during the assay. The steviol glycoside substrates Reb A, S13G, and Rubu were used in steady-state kinetic assays. Reb A or S13G respectively accepts a single β (1–2) glucose transfer at the R2 or R1 end during the assay. S13G may accept β (1–6) glucose transfer after completing β (1–2) glucose transfer. During the assay, the consumption of S13G is <10% of the initial concentration, and β (1–6) glucosylation can be ignored. Since Rubu participates in two β (1–2) sugar transfers, we cannot distinguish the production of UDP at individual ends. The kinetic data of Rubu reflect the combination of dual β (1–2) glucose transfers at both the R1 and R2 ends. All assays were performed in technical triplicate ($n = 3$). Steady-state kinetic parameters of the steviol glycoside substrate were determined by nonlinear fitting with shared values (global fitting) to all the triplicate datasets according to the Michaelis–Menten equation when UDPG was at 200 µM, which was ~10 times its $K_m$. The original data are provided as a Source Data file.

**Reporting summary.** Further information on research design is available in the Nature Research Reporting Summary linked to this article.

## Data availability
The coordinates and structure factors have been deposited in the Protein Data Bank under accession codes 7ERY [https://doi.org/10.2210/pdb7ERY/pdb] (Apo OsUGT91C1), 7ES0 [https://doi.org/10.2210/pdb7ES0/pdb] (OsUGT91C1 + UDP + Reb E), 7ES1 [https://doi.org/10.2210/pdb7ES1/pdb] (OsUGT91C1 + UDP + ST), 7ERX [https://doi.org/10.2210/pdb7ERX/pdb] (OsUGT91C1 + UDP + STB), 7ES2 [https://doi.org/10.2210/pdb7ES2/pdb] (OsUGT91C1 H27A + UDP + Reb D). The source data underlying the kinetic parameters in Table 1 are provided with this paper as a Source Data file. All relevant data generated in this study are provided in the main text, the Supplementary information, or the Source data file. Four glycosyltransferases of *S. rebaudiana* are annotated as GT1 family members in the Carbohydrate-Active enZymes (CAZy) Database by searching their GenBank accession codes: AAR06916.1, AAR06920.1, ACE87855.1, and AGL95113.1. Source data are provided with this paper.

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

## Acknowledgements

We thank the staff of the BL18U1 and BL19U1 beamline at the Shanghai Synchrotron Radiation Facility, Zhangjiang Lab, for assistance during data collection. This work was funded by grants to X.Z. from National Key Research and Development Program of China (2018YFC1002803), National Natural Science Foundation of China (31771910), and Sichuan Science and Technology Program (2021JDRC0029). J.H.N. is supported by Wellcome Trust (100209/Z/12/Z). The Rosalind Franklin Institute is a core funded research Institute of the EPSRC. W.C. is supported by National Natural Science Foundation of China (31870836), the 1.3.5 Project for Disciplines Excellence of West China Hospital, Sichuan University (ZYYC20005), and Key Science and Technology Research Projects in Key Areas of the Corps (2018AB019).

## Author contributions

X.Z., J.H.N., W.C., J. Zhang designed experiments, interpreted data, and wrote the manuscript; J. Zhang and Y.C. performed the crystallization, crystal diffraction data collection, biochemical characterization, and interpreted data; M.T. performed LC-MS and mass spectrum analyses; D.K., J. Zhou, X.X. and W.Y. performed the biochemical characterization, and crystal diffraction data collection; J.H., H.D., Y.W. and Y.L. contributed to materials, instruments, data analysis, and manuscript; X.Z. and J.H.N. solved and analyzed the X-ray crystal structures. All authors approved the final manuscript.

## Competing interests

The authors declare no competing interests.
