## [Peer Review File · Nature Communications]

Catalytic flexibility of rice glycosyltransferase OsUGT91C1 for the production of palatable steviol glycosidesREVIEWER COMMENTS

Reviewer #1 (Remarks to the Author):

This work describes a beautiful combination of biochemistry, structural biology and rational engineering to develop improved glycosylation catalysts for the production of highly desirable, but rare steviol glycosides. The increasing interest in the use of non-calorigenic and non-cariogenic sweeteners means that this is significant and important work, with potential to improve industrial production of these important items of commerce.

1. Abstract: it is stated that steviol glycoside could serve as low-calorie sweeteners. I was under the impression that they were used as a sweetener already? Add the name of the plant source to the abstract to support searching. The statement that the *Oryza* enzyme uses ‘two discrete modules’ should be reworded to ‘two discrete modes’ or perhaps at ‘two discrete subsites’?

2. Line 93. Final paragraph of the Introduction. I found it confusing that the first two sentences include background literature, while the third sentence starts to discuss the work in this manuscript, but uses the past tense (showed) rather than the present (show). Please join the first two sentences to the previous paragraph and fix the tense problem. In this same paragraph, change ‘serial’ (also at line 383) to ‘series of’, as this confuses with serial crystallography, which was not used here.

3. The authors state that this work is an exception that a GT makes two types of bond. While the one-bond, one-transferase notion is often stated, there are plenty of counter-examples. Other cases include most notably GalT-I, which changes its specificity in the presence of lactalbumin; other examples include: Green Synthesis and Catalysis 2 (2021) 45–53, Applied Microbiology and Biotechnology 2016, 100, 8411–8424, and J Biol Chem, 2006, 281, 34441–34447. Nonetheless, the present example is noteworthy.

4. Line 230. It is mentioned that the R1 glycan cannot be identified because of its flexibility. This should be reworded to say that the lack of electron density meant that the R1 glycan could not be modeled, most likely owing to disorder, which could indicate flexibility.

5. Lines 235, 244. Change ‘glycosidic ether bond’ to ‘glycosidic bond’.

6. Line 275. The sentence starting “The glycosyltransferase...” is largely repeated in the later sentence at line 290. Delete the latter one.

7. Line 271 onwards, and Supp Fig 7. The authors note that EUGT11 can cleave sugar residues in the presence of UDP, but describe this reaction as a hydrolysis (which would yield glucose). Yet the references given instead suggest a reverse glycosyl transfer reaction (which yields the sugar nucleotide). Presumably this is the case here also? A recent report on this reverse glycosyl transfer phenomenon: Cell Host Microbe, 2019, 26, 385-399. Were the authors able to confirm the formation of UDPGlc?

8. Line 292. It is stated that the reverse reaction is favored only when the donor supply is exhausted. This is not quite the case, rather it is an equilibrium reaction, and the reverse reaction is favored when there is an excess of UDP (Le Chatelier’s principle), as demonstrated by the authors’ own experiments.

9. Line 429. Sentence “EUGT11 shows dramatic catalytic plasticity...” overstates the situation. It would be better to state that “EUGT11 shows substrate plasticity...”. Also, I did not know what was meant when it was stated that “the substrate-binding site is comprehensive to manage each reaction.” Please rewrite.

10. Table 1. The accuracy of the values is overstated. Errors should be to at most two significant figures, and the corresponding data rounded. Are the errors SE or SD? The percentage term in the final column needs to be explained. The symbol for seconds is s, not S. It is confusing to have kact in min⁻¹, Km in micromolar, yet kcat/KM as per second per millimolar.

Reviewer #2 (Remarks to the Author):

I tried reading the manuscript but it was too difficult to comprehend fully. There was a lack of context regarding other UGTs. A lack of context regarding CAZY database relationships. Reject because the writing was grammatically unclear.

I do not like the format of the various steviol glycosides presented in Figure 1. The authors should use standard scientific notation for glycosidic linkages, rather than the singular numbers in each ring. Why because the numbers are not consistent. The number 1 indicates that the anomeric centre is linked to the aglycone, whereas the number 2 indicates that the anomeric centre is linked beta(1-2), whilst the 3 means the anomeric centre is linked beta(1-3) whilst 5 means it is linked beta(1-6).

Figure 1b is not clear EUGT11 is over one solid black arrow, but not another.

I would prefer the aglycone attachment sites to be called C13 and C19 in the text and figures, as this identifies a carbon atom on the aglycone backbone, rather than R1 or R2. I do not like 1R1, 1R2 or 1R6. This labelling scheme is arbitrary. Also, R is also used to depict arginine in the X-ray structure, so a double use of a letter in a figure is poor scientific literacy.

The kinetic parameters should be linked to the chemical schemes shown in figures 2 and 4, so that the rates of glycosyltransfer are visually accessible to the reader for the wild-type enzyme. I think figure 4 should be joined onto figure 2, particularly the chemical schemes as this will overall reduce the repetition of chemical structures / schemes.

Figure 5 looks like a child has drawn a cartoon bus with the righthand side site. It is shocking. Redo all three. Draw out the aglycone and carbohydrates in full, with appropriate beta(1-C19) or beta(1-6) linkage description.

Figure 1 - add the strength of sweetness beneath each structure, so that the lay reader will be better informed.

There are other examples in the literature of glycosyltransferases that add multiple two sugars. Cyclodextrin glycosyltransferases CGTase, and an enzyme from landomycin biosynthesis.

There is no situational positioning of these enzymes relative to CAZY glycosyltransferase enzymes, nor within the bank of plant glycosyltransferases - yet the authors infer the stereochemistry and mechanism of the enzyme. Are there structure of homologous glycosyltransferases?

What am I meant to understand by the following statements? Grammatically they are not clear.

L86 "We examined the cell extract of HEK293T transiently expressing UGT91D2 that only works on the addition of glucose 2R1 consistently "

L93 "UGT91D2 is not ready to express and purify in a soluble form for further study."?

L96 "and requires characterizing the use in steviol glucosylation"

L56: "A series of glycosyltransferases (UGTs) in *S. rebaudiana* utilizes uridine diphosphate-activated glucose (UDP-glucose) as the sugar donor to decorate the R1 and R2 sites with glycans through regiospecific β -glycosidic bonds (Figure 1b)" define the number of glycosyltransferases specifically. replace glycan with a more specific term, mono- di- or tri- glycosyl units

L 59: "UGT85C2 and UGT74G1 add the first glucose to the steviol aglycone by a β -configured glycosidic ether bond at the R1 site or ester bond at the R2 site, respectively"

where do these two enzymes come from, are they from *S. rebaudiana*?

A glycosidic ether bond does not make chemical sense. A glycosidic linkage is an acetal.

Sorry, but the language frustrates me such that I am not going to pour over the manuscript.

The kinetic data in the supporting information looks very consistent.

REVIEWER COMMENTS

We thank the reviewers for their constructive comments. Our responses are below in **bold blue**.

Reviewer #1 (Remarks to the Author):

This work describes a beautiful combination of biochemistry, structural biology and rational engineering to develop improved glycosylation catalysts for the production of highly desirable, but rare steviol glycosides. The increasing interest in the use of non-calorigenic and non- cariogenic sweeteners means that this is significant and important work, with potential to improve industrial production of these important items of commerce.

Answer:

Thanks for kind comments of reviewer 1.

1. Abstract: it is stated that steviol glycoside could serve as low-calorie sweeteners. I was under the impression that they were used as a sweetener already? Add the name of the plant source to the abstract to support searching. The statement that the Oryza enzyme uses ‘two discrete modules’ should be reworded to ‘two discrete modes’ or perhaps at ‘two discrete subsites’?

Answer:

Steviol glycosides are used as low-calorie sweeteners but have not yet been widely accepted. It is thought in part this is due to the taste perception (bitter taste) of the naturally abundant compounds stevioside (ST) and rebaudioside A (Reb A).

We added the plant source, *Stevia rebaudiana*, and revised ‘could serve as low-calorie sweeteners’ to ‘could widely serve as a low-calorie sweetener’.

The first sentence of the abstract now is

*‘Steviol glycosides are the intensely sweet plant extracts of *Stevia rebaudiana*. These molecules comprise an invariant steviol aglycone decorated with variable glycans and could widely serve as a low-calorie sweetener.’*

We rewrote the abstract and removed the phrase ‘two discrete modules’.

2.Line 93. Final paragraph of the Introduction. I found it confusing that the first two sentences include background literature, while the third sentence starts to discuss the work in this manuscript, but uses the past tense (showed) rather than the present (show). Please join the first two sentences to the previous paragraph and fix the tense problem. In this same paragraph, change ‘serial’ (also at line 383) to ‘series of’, as this confuses with serial crystallography, which was not used here.

Answer:

The last paragraph of the introduction has been reworded. Changed the tense to the present.

We removed the phrase ‘serial’ throughout the manuscript

3. The authors state that this work is an exception that a GT makes two types of bond. While the one-bond, one-transferase notion is often stated, there are plenty of counter-examples. Other cases include most notably GalT-I, which changes its specificity in the presence of lactalbumin; other examples include: Green Synthesis and Catalysis 2 (2021) 45–53, Applied Microbiology and Biotechnology 2016, 100, 8411–8424, and J Biol Chem, 2006, 281, 34441– 34447. Nonetheless, the present example is noteworthy.

Answer:

The paragraph has been reworded. We thank the reviewer for pointing out these examples. The statement of ‘the exception’ is removed, and citation to other examples is made in Discussion.

4. Line 230. It is mentioned that the R1 glycan cannot be identified because of its flexibility. This should be reworded to say that the lack of electron density meant that the R1 glycan could not be modeled, most likely owing to disorder, which could indicate flexibility.

Answer:

Changed to “*The R1 glycan of ST was not modeled into the weak experimental electron density observed at this location, indicative of disorder due to flexibility*”

5. Lines 235, 244. Change ‘glycosidic ether bond’ to ‘glycosidic bond’.

Answer:

We have corrected this and checked throughout the manuscript. Apologies.

6. Line 275. The sentence starting “The glycosyltransferase...” is largely repeated in the later sentence at line 290. Delete the latter one.

Answer:

We have deleted this.

7. Line 271 onwards, and Supp Fig 7. The authors note that EUGT11 can cleave sugar residues in the presence of UDP, but describe this reaction as a hydrolysis (which would yield glucose). Yet the references given instead suggest a reverse glycosyl transfer reaction (which yields the sugar nucleotide). Presumably this is the case here also? A recent report on this reverse glycosyl transfer phenomenon: Cell Host Microbe, 2019, 26, 385-399. Were the authors able to confirm the formation of UDPGlc?

Answer:

We were always considering this reaction is the reverse glycosyl transfer back to UDP and apologize for the misuse of the phrase ‘hydrolysis’. We thank the reviewer for pointing out the recent report and cited ‘Cell Host Microbe, 2019, 26, 385-399’.

We didn't confirm the formation of UDP-Glucose directly, but judged it based on two indirect observations:

- 1) The β (1-2) sugar cleavage of steviol compound is UDP-dependent. As some glycosyltransferases have been reported to be reversible, we suspect this is the reverse reaction, transferring sugar from steviol compound, such as Reb E, ST and STB, etc. back to UDP and yield UDP-glucose.**
- 2) In the crystal structure of the enzyme with Reb E, only excess UDP was present. Reb E (containing glucose 1-R1, 2-1-R1, 1-R2, 2-1-R2) (apologize for changing glucose nomenclature) was converted to a new compound (1-R1, 2-1-R1, 6-1-R1, 1-R2) during the crystallization. The structure showed that glucose 2-1-R2 was removed and glucose 6-1-R1 was added (enzyme-catalyzed sugar exchange). This is consistent with production of UDP-glucose as the intermediate. We did not include this in the manuscript since the explanation is complex (in an already complex story) and any such chemistry inside crystallization experiments may not be fully reflective of the solution reactions. However, it indeed inspired us to examine that the enzyme catalyzes the cleavage of β (1-2) sugar and β (1-6) glucosylation individually.**

8. Line 292. It is stated that the reverse reaction is favored only when the donor supply is exhausted. This is not quite the case, rather it is an equilibrium reaction, and the reverse reaction is favored when there is an excess of UDP (Le Chatalier's principle), as demonstrated by the authors' own experiments.

Answer:

We apologize the confusion and this has been reworded.

The reviewer is right about the equilibrium of the reversible reactions. As UDP-glucose formation was thermodynamically disfavored and typically required at least excess of UDP for sugar nucleotide production. We can't say 'the reverse reaction is favored only when the donor supply is exhausted', because it is not the case when donor supply is exhausted if it is not excess.

9. Line 429. Sentence "EUGT11 shows dramatic catalytic plasticity..." overstates the situation. It would be better to state that "EUGT11 shows substrate plasticity...". Also, I did not know what was meant when it was stated that "the substrate-binding site is comprehensive to manage each reaction." Please rewrite.

Answer:

We have reworded as

'OsUGT91C1 shows unusual catalytic promiscuity in that it catalyzes three distinct glucosylation reactions at the same active site, β (1-2) to the R1 end and R2 end, β (1-6) to the R1 end.'

10. Table 1. The accuracy of the values is overstated. Errors should be to at most two significant figures, and the corresponding data rounded. Are the errors SE or SD? The percentage term in the final column needs to be explained. The symbol for seconds is s, not

S. It is confusing to have kact in min⁻¹, Km in micromolar, yet kcat/KM as per second per millimolar.

Answer:

We have changed the significant figures of the errors to 2 and rounded the corresponding data.

The kinetic data were obtained from the nonlinear regression. The fitting program 'Graphpad prism' is using "Std. Error" or SE, and stating that "standard error" and "standard deviation" are the same meaning for the fitting parameters from regression.

The standard error of a parameter is the expected value of the standard deviation of that parameter if the experiment was repeated many times. We did the assays in technical triplicate (n=3). We prefer to call the errors SD and added a note to Table 1,

'Nonlinear fitting values \pm S.D. (n=3) are shown'.

We changed percentage term in table to a separate column 'fold', that shows fold change over k_{cat}/K_m of the wide type.

We changed the second symbol to low case, which was a typo from autocorrection.

We used min^{-1} as the unit of k_{cat} , μM for K_m , the custom unit for kinetic parameter, while we chose the standard unit of ' $\text{s}^{-1} \text{M}^{-1}$ ' for k_{cat}/K_m for convenient comparison of catalytic efficiency between enzymes.

Reviewer #2 (Remarks to the Author):

I tried reading the manuscript but it was too difficult to comprehend fully. There was a lack of context regarding other UGTs. A lack of context regarding CAZY database relationships. Reject because the writing was grammatically unclear.

Answer:

We apologize for the quality of the presentation and have rewritten the manuscript.

We have now put in the context of the CAZY database. We have changed EUGT11 to OsUGT91C1 (UGT91C1, NCBI Reference Sequence: XP_015629141.1).

I do not like the format of the various steviol glycosides presented in Figure 1. The authors should use standard scientific notation for glycosidic linkages, rather than the singular numbers in each ring. Why because the numbers are not consistent. The number 1 indicates that the anomeric centre is linked to the aglycone, whereas the number 2 indicates that the anomeric centre is linked beta(1-2), whilst the 3 means the anomeric centre is linked beta(1-3) whilst 5 means it is linked beta(1-6).

Answer:

We did use the numbers consistently. We did use '6', instead of '5', to represent the linkage of beta (1-6).

We apologize for the lack of clarity. It is quite a complex system that soon becomes clumsy in our hands if using the normal nomenclature.

We have tried to improve this.

We found it does help to name the sugars by some short name and at the same time, used Figure to clearly set out those short names.

Figure 1b is not clear EUGT11 is over one solid black arrow, but not another.

Answer:

We have added an explanation in legend of Figure 1b.

"The reactions involving OsUGT91C1 or UGT91D2 are boxed in light blue. UGT91D2 adds glucose 2-1-R1 but shows only trace catalytic activity for addition of glucose 2-1-R2 (denoted with a dashed arrow). OsUGT91C1, studied here, efficiently adds of both glucose 2-1-R2 and glucose 2-1-R1 (red arrows)."

I would prefer the aglycone attachment sites to be called C13 and C19 in the text and figures, as this identifies a carbon atom on the aglycone backbone, rather than R1 or R2.

Answer:

The R1 and R2 nomenclature has been used previously and we found that calling back to C13 or C19 through the text did not help clarity. However, in Figure 1 and the text we have clearly spelt out the attachment to C13 for R1 and C19 for R2.

I do not like 1R1, 1R2 or 1R6. These labelling scheme is arbitrary. Also, R is also used to depict arginine in the X-ray structure, so a double use of a letter in a figure is poor scientific literacy.

Answer:

We note the reviewer's objection and have tried to define a clearer

nomenclature: Glucose 1-R1, represents the glucose that is directly attached to C13 (R1) of the aglycone, i.e. the first glucose at the R1 end of aglycone.

Glucose 1-R2, represents the glucose that is directly attached to C19 (R2) of the aglycone, i.e. the first glucose at the R2 end of aglycone.

As Glucose 1-R1 and 1-R2 provide acceptor site for subsequent glucosylation, we used the following short names.

Glucose 2-1-R1, represents the glucose that is linked to the 2-hydroxyl of glucose 1-R1. Glucose 3-1-R1, represents the glucose that is linked to the 3-hydroxyl of glucose 1-R1. Glucose 6-1-R1, represents the glucose that is linked to the 6-hydroxyl of glucose 1-R1.

...

We think using the form we have now, the reader is less likely to confuse with arginine.

The kinetic parameters should be linked to the chemical schemes shown in figures 2 and 4, so that the rates of glycosyltransfer are visually accessible to the reader for the wild-type enzyme. I think figure 4 should be joined onto figure 2, particularly the chemical schemes as this will overall reduce the repetition of chemical structures / schemes.

Answer:

The kinetic data were measured in a different method from Fig 2 and 4, also they are in the different time scale (min vs h). It is not helpful to combine them together.

Figure 2 establishes the promiscuity of the native enzyme for the β (1-2) sugar transfer. Figure 4 shows the data for the β (1-6) sugar transfer. The ability to do both β (1-2) and β (1-6) is surprising and we felt β (1-6) deserved its own figure for evidence of this behavior.

Also, the promiscuity of β (1-6) is different in origin from that of β (1-2). The discussion of the different promiscuities arose at different points in the manuscript and therefore we prefer to keep separate figures.

The chemical cartoons we think help with the understanding.

Figure 5 looks like a child has drawn a cartoon bus with the righthand side site. It is shocking. Redo all three. Draw out the aglycone and carbohydrates in full, with appropriate beta(1-C19) or beta(1-6) linkage description.

Answer:

Fig. 5 has been redrawn. The aglycone and carbohydrates are exactly in full.

Figure 1 - add the strength of sweetness beneath each structure, so that the lay reader will be better informed.

Answer:

Where known these are included in Fig 1.

There are other examples in the literature of glycosyltransferases that add multiple two sugars. Cyclodextrin glycosyltransferases CGTase, and an enzyme from landomycin biosynthesis.

Answer:

We have made clear we do not claim OsUGT91C1 is “unique” but it is unusual and have cited other examples.

Cyclodextrin glycosyltransferases CGTase, and an enzyme from landomycin biosynthesis are more likely to catalyze the linear extension of the glycans and different from the enzyme we are working on.

There is no situational positioning of these enzymes relative to CAZY glycosyltransferase enzymes, nor within the bank of plant glycosyltransferases - yet the authors infer the stereochemistry and mechanism of the enzyme. Are there structure of homologous glycosyltransferases?

Answer:

The stereochemistry of the reaction of OsUGT91C1 flows from the identification of authentic standards. The mechanism is one of the catalytic motifs employed by the inverting GT-B glycosyltransferase. We have cited the previous studies on the mechanism of this class.

We did not probe the mechanism here, rather we rely only on identification of His27 as the base to decipher the structural basis for the usual promiscuity.

There are not available closely homolog structures to OsUGT91C1 and we solved the structure by SAD.

What am I meant to understand by the following statements? Grammatically they are not clear.

L86 "We examined the cell extract of HEK293T transiently expressing UGT91D2 that only works on the addition of glucose 2R1 consistently "

Answer:

We rewrote the manuscript and deleted this sentence.

L93 "UGT91D2 is not ready to express and purify in a soluble form for further study."?

Answer:

We rewrote the manuscript and deleted this sentence.

L96 "and requires characterizing the use in steviol glycosylation"

Answer:

We rewrote the manuscript and deleted this sentence.

L56: "A series of glycosyltransferases (UGTs) in *S. rebaudiana* utilizes uridine diphosphate- activated glucose (UDP-glucose) as the sugar donor to decorate the R1 and R2 sites with glycans through regiospecific β -glycosidic bonds (Figure 1b)" define the number of glycosyltransferases specifically. replace glycan with a more specific term, mono- di- or tri- glycosyl units

Answer:

Changed to 'The glucosylation of steviol species relies on four glycosyltransferases (UGTs) (Fig. 1 b) located in the cytosol of *S. rebaudiana*'.

The glycosyltransferases are mixed in the cytosol of *S. rebaudiana*, and some enzymes operate at both the R1 and R2 ends, others only at one end. Therefore, the number of the glucose moieties vary from 1 to 3 at the R1 and R2 ends and give rise to a stochastic population. We don't find a proper way to describe the specific number of the glucose units.

L 59: "UGT85C2 and UGT74G1 add the first glucose to the steviol aglycone by a β -configured glycosidic ether bond at the R1 site or ester bond at the R2 site, respectively"

where do these two enzymes come from, are they from *S. rebaudiana*?

Answer:

Yes, they are and we have now stated.

A glycosidic ether bond does not make chemical sense. A glycosidic linkage is an acetal.

Answer:

Corrected.

Sorry, but the language frustrates me such that I am not going to pour over the manuscript.

Answer:

We have rewritten the manuscript.

The kinetic data in the supporting information looks very consistent.

We are confident that the kinetic analyses were performed properly. All experimental data recorded by the plate reader are included in source data file.

In detail

'Steady-state kinetic assays of those substrates were performed using the UDP-Glo™ Glycosyltransferase Assay Kit (Promega). The sugar transfer of the glycosyltransferase turns UDP-glucose to UDP, which is converted to ATP and then quantified using a luciferase/luciferin reaction. The luminescence intensity linearly correlates to the concentration of UDP when it is less than 25 μ M, thus reflecting UDP production during the reaction. A standard linear plot between the luminescence intensity and UDP concentrations was drawn for each microplate measurement (R2 > 0.98).

After the reaction was started by adding the enzyme, four individual aliquots were taken every 1 min and mixed with an equal volume of the UDP detection reagent, terminating the reaction and quantifying the velocity of UDP production. The blank turnover of UDPG to UDP in the absence of the steviol substrate was subtracted from each velocity calculation. The appropriate concentrations of the enzyme were tested to ensure that the consumption of both steviol substrate and UDPG was <10% of the initial concentration during the assay.

All assays were performed in technical triplicate. Steady-state kinetic parameters of the steviol substrate were determined by nonlinear fitting according to the Michaelis–Menten equation when UDPG was at 200 μM , which was approximately 10 times its K_m .

During the plate reading, the instrument was set to record values with 4 significant digits in scientific notation. When exported as normal digits, this adds zeros. We wondered if this lay behind the comment about consistency? All data are presented in scientific notation in the revision.

REVIEWERS' COMMENTS

Reviewer #1 (Remarks to the Author):

The deep rewriting of the entire manuscript and the reconstruction of the structural figures and transfer of some parts to the SI has dramatically enhanced the readability of the manuscript and the clarity of the figures. I am satisfied that the authors have tried hard to accommodate all the reasonable requests of the reviewers.

1. Remove cyclodextrin glycosyltransferase as an example. It is classified as a glycoside hydrolase (family GH13) and does not utilize sugar nucleotide substrates
2. Delete "HPLC traces in" at start of Figure 2a legend.

Reviewer #3 (Remarks to the Author):

The manuscript details biochemical and structural characterization of glycosyltransferase OsUGT91C1 catalyzing glucosylation of several steviol glycosides. Among the key results is the catalytic promiscuity of this enzyme: OsUGT91C1 is able to form β (1-2) glycosidic linkages at two distinct acceptor sites of steviol glycosides, with glucose 1-R1 at C13 and glucose 1-R2 at C19, and β (1-6) glycosidic linkage with glucose 1-R1 at C13. Four co-crystal structures (and an apo structure) are reported with four different steviol glycosides which not only allow credible rationalization of such catalytic promiscuity but were also the basis for subsequent rational enzyme engineering. Point mutations of the binding site were designed and tested, and ultimately resulted in eliminated undesirable glucosylation (β (1-6) at Glc 1-R1) and impressively \sim 4-fold improved highly desirable glucosylation (β (1-2) at Glc 1-R2). Although OsUGT91C1 and its activity on Glc 1-R2 have been reported previously (as EUGT11), this work adds significantly to prior knowledge.

There are practical applications of this work as zero-calorie sweeteners have received a lot of attention recently due to their potential health benefits in replacing sugar. At the same time, generation of advanced better testing steviol glycosides, such as Reb M, at scale and in pure form is still a challenge. Specifically, highly efficient enzyme for β (1-2) glucosylation at Glc 1-R2 is highly desirable. Authors contribute to advancement in this area by providing means to rationally engineer a glycosyltransferase activity necessary for Reb M biosynthesis. In addition to practical applications, this work is valuable in understanding the substrate specificity and catalytic promiscuity of glycosyltransferases (parallel with UGT76G1 was a great addition to Discussion). Another aspect, reversibility of glycosyltransferases, has

been previously documented but still rarely reported and this manuscript provides an interesting example.

The insights and conclusions expressed in the manuscript are well supported by the wealth of biochemical and structural analysis as well as success of rational engineering based on these analyses. The experiments are well thought out, well executed, and described in sufficient detail to be reproduced.

Generally, figures were clear and informative. I appreciated Figure 1 as it contained a lot of information clearly presented using chemical structures, cartoons, colors, etc. As one of the previous reviewers, I also found that using nomenclature "R1, R2" as opposed to "C13, C19" was difficult to get used to initially, but not opposed to keeping as is since it is clearly explained.

I would like to suggest few minor modifications to help improve clarity of the manuscript:

Lines 29-30 (abstract): Replace "steviol substrate" with "steviol glycoside substrates" (steviol is not a substrate of this UGT, but various steviol glycosides are)

Lines 71-72: Sentence "In *S. rebaudiana*, the enzyme UGT85C2 adds the first glucose to the R1 end through an acetal linkage, while UGT74G1 adds the first glucose to the R2 end through an ester linkage (Fig. 1 b)." gave me a pause as these are both glycosidic linkages. Since chemical nature of accepting sites C13 vs C19 is not the point of discussion, I would modify for clarity to "In *S. rebaudiana*, the enzyme UGT85C2 adds the first glucose to C13 hydroxyl group (R1 end), while UGT74G1 adds the first glucose to C19 carboxylate moiety (R2 end) (Fig. 1 b)."

Line 422: use superscript in "ml⁻¹"

The supplementary figures 5 (b, c), 7 (e), 8 (b), 9 (b) still contain EUGT11 labels for mutants whereas OsUGT91C1 is used throughout manuscript. Please change to mutation only as in Figure 10.

REVIEWERS' COMMENTS

We thank the reviewers for their constructive comments. Our responses are below in bold blue.

Reviewer #1 (Remarks to the Author):

The deep rewriting of the entire manuscript and the reconstruction of the structural figures and transfer of some parts to the SI has dramatically enhanced the readability of the manuscript and the clarity of the figures. I am satisfied that the authors have tried hard to accommodate all the reasonable requests of the reviewers.

1. Remove cyclodextrin glycosyltransferase as an example. It is classified as a glycoside hydrolase (family GH13) and does not utilize sugar nucleotide substrates

Answer:

Thanks for pointing this out. We have deleted this example and the corresponding reference.

2. Delete "HPLC traces in" at start of Figure 2a legend.

Answer:

We have deleted 'HPLC traces in' in Figure 2a, Figure 4, and supplementary figures.

Reviewer #3 (Remarks to the Author):

The manuscript details biochemical and structural characterization of glycosyltransferase OsUGT91C1 catalyzing glucosylation of several steviol glycosides. Among the key results is the catalytic promiscuity of this enzyme: OsUGT91C1 is able to form β (1-2) glycosidic linkages at two distinct acceptor sites of steviol glycosides, with glucose 1-R1 at C13 and glucose 1-R2 at C19, and β (1-6) glycosidic linkage with glucose 1-R1 at C13. Four co-crystal structures (and an apo structure) are reported with four different steviol glycosides which not only allow credible rationalization of such catalytic promiscuity but were also the basis for subsequent rational enzyme engineering. Point mutations of the binding site were designed and tested, and ultimately resulted in eliminated undesirable glucosylation (β (1-6) at Glc 1-R1) and impressively \sim 4-fold improved highly desirable glucosylation (β (1-2) at Glc 1-R2). Although OsUGT91C1 and its activity on Glc 1-R2 have been reported previously (as EUGT11), this work adds significantly to prior knowledge.

There are practical applications of this work as zero-calorie sweeteners have received a lot of attention recently due to their potential health benefits in replacing sugar. At the same time, generation of advanced better tasting steviol glycosides, such as Reb M, at scale and in pure form is still a challenge. Specifically, highly efficient enzyme for β (1-2) glucosylation at Glc 1-R2 is highly desirable. Authors contribute to advancement in this area by providing means to rationally engineer a glycosyltransferase activity necessary for Reb M biosynthesis. In addition to practical applications, this work is valuable in understanding the substrate specificity and catalytic promiscuity of glycosyltransferases (parallel with UGT76G1 was a

great addition to Discussion). Another aspect, reversibility of glycosyltransferases, has been previously documented but still rarely reported and this manuscript provides an interesting example.

The insights and conclusions expressed in the manuscript are well supported by the wealth of biochemical and structural analysis as well as success of rational engineering based on these analyses. The experiments are well thought out, well executed, and described in sufficient detail to be reproduced.

Generally, figures were clear and informative. I appreciated Figure 1 as it contained a lot of information clearly presented using chemical structures, cartoons, colors, etc. As one of the previous reviewers, I also found that using nomenclature “R1, R2” as opposed to “C13, C19” was difficult to get used to initially, but not opposed to keeping as is since it is clearly explained.

I would like to suggest few minor modifications to help improve clarity of the manuscript:

Lines 29-30 (abstract): Replace “steviol substrate” with “steviol glycoside substrates” (steviol is not a substrate of this UGT, but various steviol glycosides are)

Answer:

Thanks for pointing this out. We have changed ‘steviol substrate’ with ‘steviol glycoside substrates’ for a more precise definition in the abstract and the main text.

Lines 71-72: Sentence “In *S. rebaudiana*, the enzyme UGT85C2 adds the first glucose to the R1 end through an acetal linkage, while UGT74G1 adds the first glucose to the R2 end through an ester linkage (Fig. 1 b).” gave me a pause as these are both glycosidic linkages. Since chemical nature of accepting sites C13 vs C19 is not the point of discussion, I would modify for clarity to “In *S. rebaudiana*, the enzyme UGT85C2 adds the first glucose to C13 hydroxyl group (R1 end), while UGT74G1 adds the first glucose to C19 carboxylate moiety (R2 end) (Fig. 1 b).”

Answer:

We have changed as the reviewer suggested.

Line 422: use superscript in “ml⁻¹”

Answer:

We have corrected and used superscript in ‘ml⁻¹’.

The supplementary figures 5 (b, c), 7 (e), 8 (b), 9 (b) still contain EUGT11 labels for mutants whereas OsUGT91C1 is used throughout manuscript. Please change to mutation only as in Figure 10.

Answer:

We have corrected this. Apologies.